# Marital Status, Father Acknowledgement, and Birth Outcomes: Does the Maternal Education Matter?

**DOI:** 10.3390/ijerph20064868

**Published:** 2023-03-10

**Authors:** Anna Merklinger-Gruchala, Maria Kapiszewska

**Affiliations:** Faculty of Medicine and Health Sciences, Andrzej Frycz Modrzewski Krakow University, 30-705 Krakow, Poland

**Keywords:** birth weight, birth outcomes, paternal acknowledgement, father absence, social support, educational status, marital status, family structure, psychological stress, non-traditionally married mothers

## Abstract

We evaluated whether the maternal marital status and father acknowledgement (proxy for paternal presence) affect birth weight, and if so, whether the maternal educational attainment modifies this effect. The growing tendency of alternative forms of family structure affects maternal well-being and pregnancy outcome. However, it is not known whether poorer birth outcomes of out-of-wedlock childbearing can be overcome or compensated by maternal education. Using birth registry data, we assessed the impact of maternal civil status and child recognition by the father on birth-weight-for-gestational age (BWGA) z-scores, with respect to maternal education, among Polish mothers (N = 53,528). After standardization, the effect of being unmarried with father acknowledgement (UM-F) vs. married with father acknowledgement (M-F) reduced the BWGA z-score of 0.05 (*p* < 0.001), irrespective of educational attainment (*p* for interaction = 0.79). However, education differentiated the effect of father acknowledgement across unmarried mothers. BWGA z-scores were significantly lower among the low-educated unmarried group without father acknowledgment (UM-NF) as compared to UM-F (equaled −0.11, *p* = 0.01). The same effect among the higher-educated group was non-significant (*p* = 0.72). Higher maternal education can compensate for the negative effect of a lack of father acknowledgement, but it does not help to overcome the effect of out-of-wedlock stress exposure.

## 1. Introduction

In the traditional, patriarchal model of family, the father’s presence and a trusting relationship between the father and pregnant mother contribute to a mother’s sense of security and ensure access to financial resources. This traditional model has many pros and cons and has lasted for many generations, but in recent decades it has slowly and gradually been turning into a democratic contemporary family as couples voluntarily enter into marriage or partners choose to live together without being legally married to each other [1]. The growing tendency of these alternative forms of family structure is generally identified as cohabitation and raises the question in regard to childbearing about the implications of this trend on the well-being of offspring. It cannot be ignored that an extensive body of research demonstrates that in comparison to marriage, cohabitation, despite its acceptability as a social institution, is associated with lower relationship quality and commitment and higher rates of union instability [2,3]. Women in such relationships are more likely to suffer from a perceived lack of commitment from their partners than their married counterparts, and a child born outside of marriage is still categorized in some places as illegitimate [4]. This risk is higher in countries where the social norm with respect to female gender roles is defined in more traditional terms, and countries with conservative religious norms [5]. In addition, in more conservative countries, cohabiting couples experience different judicial outcomes in relation to social security and tax, inheritance rights, and welfare benefits, as compared to married couples. The lower levels of well-being characterizing cohabitors and the uncertainty and insecurity in the relationship (higher among unmarried than married couples) seem to be mostly responsible for maternal distress during pregnancy [6] that can negatively influence fetal growth as measured by birth weight [3].

Stress may be particularly devastating when the father does not acknowledge paternity, indicated by the lack of the father’s name on the birth certificate [7,8,9,10]. As authors suggest, the relationship between the mother and father in these circumstances is of short duration and characterized by low paternal confidence [10], or the mother does not know who the father of her child is or does not want to identify him for many different reasons. The risks of low birth weight (LBW) and preterm birth are higher among unmarried mothers without paternal involvement compared to married couples, as has been previously reported [2,11]. Thus, an informal relationship with a partner, unwanted pregnancies, and lack of paternity establishment [12] increase the vulnerability of pregnant mothers to prenatal stress correlating with poor birth outcomes [13,14,15]. Unfortunately, none of these results emphasize the importance of the maternal relationship status (e.g., marriage, cohabitation, or being single) and the legal status of the child as a way to mitigate this problem.

In our work, we hypothesize that one of the factors that can increase stress resilience in vulnerable groups of pregnant women is an enhanced maternal socioeconomic status as determined by economic, occupational, and educational achievements [16,17]. A higher socioeconomic status increases the level of satisfaction and happiness in both marriage and cohabitation unions [18]. Moreover, maternal educational attainment seems to strengthen the sense of security and self-esteem of pregnant women [19], thus helping to overcome the stigma associated with single motherhood and social barriers [20]. In addition, women with a higher education may not feel the pressure of the economic advantages of marriage, which is also likely to buffer against prenatal distress [21]. For example, higher maternal education is associated with a lower risk of adverse perinatal outcomes compared to low education attainment [16,22].

Given this knowledge, we aimed to determine whether maternal education can overcome or compensate for the adverse neonatal conditions (proxied by birth-weight-for-gestational age (BWGA) z-scores) associated with a mother’s civil status and the acknowledgment of paternity by the father.

## 2. Participants, Ethics, and Methods

The study group comprised the singleton live-born infants born at term (between 37 and 42 weeks of gestation) to mothers whose residence at the time of the infant’s birth was the city of Krakow. The anonymized data from the birth registry (Central Statistical Office in Poland) included birth records from 1 October 1995 to December 2009, which comprised the following: month and year of birth, birth weight (in grams), infant’s sex, maternal age (in years), gestational age (in weeks), parity, maternal education (primary, lower secondary, basic vocational, upper general, or specialized secondary and academic education), maternal employment status (employed vs. unemployed), and maternal marital status (married vs. unmarried, i.e., single, widowed, divorced, or separated). Paternal data, if present, included age (in years), education (primary, lower secondary, basic vocational, upper general, or specialized secondary and academic education), and employment status (employed vs. unemployed).

Paternal and maternal education (EDU) was divided into two levels: higher education (passed at least final high school exams, i.e., upper general or specialized secondary and academic education) and lower education (secondary education without final high school exams, i.e., primary education, lower secondary, or basic vocational education).

The maternal marital status and father acknowledgement (called the Index) were defined as the categorical variable with three levels: (1) married—a legal father of the child (according to the Polish law, the mother’s husband is automatically considered to be the father of the child) (M-F); (2) unmarried—a father voluntarily signs a paternity statement (UM-F); and (3) unmarried—a lack of father acknowledgement of a child on the birth certificate (UM-NF). Father acknowledgement of a child was proxied by the presence or absence of paternal data (age, education, and employment) on the birth record. The presence of these data indicated the voluntarily acknowledgement of paternity or court-established fatherhood. It is likely that in such circumstances the father is more prone to be in a close relationship with the mother and be involved in taking care of her during pregnancy and thus support her, forming the cohabiting union. Conversely, the lack of a father’s data on the birth record (i.e., when all fields provided for the father’s data, his age, education, and employment, were blank) may indicate that the mother does not know the father or is unwilling to identify him, which may indicate single motherhood. The outcome variable was assessed as the birth-weight-for-gestational age (BWGA) z-scores. We used the formula: z = (observed birth weight—mean birth weight)/SD), where the mean birth weight and SD were based on published population-based standards, and stratified by infant gender and gestational age in the completed weeks [23]. A positive z-score represented an infant with a larger BWGA than the average BWGA of an infant in the reference population (faster fetal growth), whereas a negative z-score denoted an infant with a BWGA smaller than the reference population (slower fetal growth). For example, a z-score of −1 represented an individual birth weight that was 1 standard deviation below the mean birth weight of the reference cohort of a given sex and gestational age. We used Fenton growth charts as the reference because they are the most popular newborn growth charts worldwide, and they are also the charts used by Polish neonatologists [24]. Using Fenton z-scores not only enabled making comparisons of neonatal birth weight within the study group, but also facilitated harmonization across the different cohorts, including those from different countries [23].

Of 88,474 singleton births, we excluded n = 4725 born before 37 and after 42 completed weeks of gestation; n = 129 stillbirths; n = 3794 births categorized as “large-for-gestational age” (LGA, according to Fenton growth charts, that is >90 percentile); and n = 9134 births categorized as “small-for-gestational age” (SGA, according to Fenton growth charts, that is <10 percentile). The intent of these exclusions was to conduct the study on singleton, live, full-term births and estimate the differences in the rate of fetal growth within a range of appropriate weights for gestational age. Mothers below 25 years of age at delivery (n = 17,166) were also excluded. The education levels of the mothers’ study groups were defined according to the International Standard Classification of Education (ISCED) and were restricted to mothers at least 25 years and above in order to ensure that the majority of the population had completed their education. These exclusions left N = 53,528 eligible births for study.

## 3. Statistical Analyses

First, we compared the characteristics of the study group stratified by maternal marital status and father acknowledgement, on the one hand, and maternal education, on the other hand, using chi-square tests for categorical variables, and *t*-tests or ANOVA for continuous variables. Then, we used a multiple linear regression model with the BWGA z-score as the dependent variable, and the index of maternal marital status and father acknowledgement (M-F, UM-F, and UM-NF), maternal education (lower/higher), infant’s sex (girl/boy), parity (primiparous/multiparous), maternal age (continuous, centered), and birth year (continuous, centered), as the independent variables. This model allowed us to estimate the main effects of this Index and EDU. Afterward, we added the product term (Index x EDU) into the multiple linear regression model as a way to test for interactions. Using this model with interaction, we also estimated simple effects of the Index of maternal marital status and father acknowledgement at a particular level of education (lower/higher) when other variables were held fixed at 0. The interplay between two factors (Index and EDU) and their influence on an outcome (BWGA z-score) was visualized. All analyses were conducted with Statistica version 13 (TIBCO Software Inc., StatSoft, Krakow, Poland).

## 4. Results

All infants were born between 37 and 42 weeks of gestation. Median birth weight was 3440 g (Q1–Q3: 3200–3700 g). Half of the study mothers were less than 30 years of age (Q1–Q3: 27–33 years), with a minimum age of 25 and a maximum age of 54, whilst the median paternal age was 31 years (Q1–Q3: 28–35 years). Among the study group, 54.1% of women were multiparous with 2–15 children. A vast majority of women were married (91%) at the time of delivery. The study group comprised more employed (82%) than unemployed women. Higher educational attainment was much more frequent among mothers than the lower one (85% vs. 15%). About 75% of fathers had a higher educational background and were employed (90%).

For n = 951 infants (1.8%), no paternal data, such as his age, employment, and education were found. This group was classified as UM-NF. The group of UM-F comprised 7.5% infants, whilst the rest of them (90.7%) were categorized as M-F.

The three levels of the Index mentioned above differed according to maternal age, maternal employment and educational levels, parity, and birth year, but not the sex of the child (Table 1). However, the groups of maternal education varied according to paternal education, maternal and paternal age, maternal and paternal employment status, Index levels, parity, and birth year, but not the sex of the child (Table 2).

We found important educational differences (*p* < 0.01) within the Index levels: women with a higher educational background more frequently gave birth while being married than unmarried: 92% M-F vs. 8% unmarried (UM-F + UM-NF). Whilst among women with a lower education, we found 84% married (M-F) vs. 16% unmarried (UM-F + UM-NF); Table 2.

Mothers with a lower education were more frequently multiparous (*p* <0.01), younger (*p* = 0.03), and with a lower percentage of being employed (*p* < 0.01) than mothers with a higher education. The paternal age of infants born by mothers with a lower vs. higher education was elevated (*p* < 0.01). Paternal unemployment was more common among lower educated vs. higher educated mothers (*p* < 0.01). The same educational levels for both mothers and fathers were found less frequently among lower-educated than among higher-educated parents (*p* < 0.01); Table 2.

An infant’s sex ratio (the ratio of boys to girls) differed according to the Index groups, with the lowest found among UM-NF (*p* < 0.01); Table 1. This UM-NF group was also found to have the highest proportion of primiparous to multiparous mothers (*p* < 0.01), and the highest proportion of unemployed mothers (*p* < 0.01).

After descriptive statistics and group comparisons, we assessed the effect of the Index levels on the BWGA z-score. After standardization to infant’s sex, year of birth, parity, maternal employment, education, and age, the mean BWGA z-score of the UM-F was −0.05 below that of the M-F (*p* < 0.01), whilst the mean BWGA z-score of the UM-NF was −0.10 below that of the M-F (*p* < 0.01). The difference of the mean BWGA z-scores between UM-F and UM-NF groups equaled −0.05 (*p* = 0.04).

Then, we assessed the effect of maternal education on the BWGA z-score. After standardization to infant’s sex, year of birth, parity, maternal employment, Index levels, and age, the mean BWGA z-score of the higher EDU was 0.07 above that of the lower EDU (*p* < 0.01).

An analysis of interaction between the Index levels and EDU, using a multiple linear regression model, indicated that the effect of Index levels varied across categories of education (Table 3). Among lower EDU mothers, while controlling for infant’s sex, parity, maternal employment, maternal age, and year of birth, the mean BWGA z-score of the UM-F was −0.05 below that of the M-F (*p* = 0.01), whilst the mean BWGA z-score of the UM-NF was −0.16 below that of the M-F (*p* < 0.01). While comparing UM-F with UM-NF mothers with lower EDU, the difference of the mean BWGA z-scores equaled −0.11 (*p* = 0.01).

Among higher EDU mothers, the difference in mean BWGA z-scores between UM-F and M-F was also 0.05 (*p* < 0.001), and it was not significantly different from the same comparison among lower EDU (*p* for interaction = 0.79).

On the other hand, the difference between UM-NF and M-F among higher EDU was not −0.16, as in lower EDU, but was significantly reduced. This reduction amounted to 0.10 (*p* for interaction 0.02). Therefore, the total difference between UM-NF and M-F among higher EDU mothers was −0.06 (−0.16 + 0.10).

The difference in mean BWGA z-scores between UM-NF and UM-F among higher EDU mothers amounted to −0.01 (*p* = 0.72), which was not statistically significant per se, but significantly smaller in comparison to the same difference found in lower EDU mothers, which equaled −0.11 (*p* for interaction = 0.045). This stronger effect of the Index of maternal marital status and father acknowledgement for lower EDU mothers is also visible in Figure 1, in which the downward BWGA z-score trend runs steepest for this group. This suggests a synergistic interaction, in which lower levels of education enhance the effect of being unmarried with no father acknowledgement on an outcome.

## 5. Discussion

We found that BWGA z-scores were significantly associated with civil status as well as educational attainment. However, educational levels did not mitigate the effects of an unmarried status on birth weight. The strong negative effects of non-marital status on birth outcomes equally affected children of lower- and higher-educated women. Therefore, there must be something more substantial about the institution of marriage per se that protects fetal growth, as Wilcox and DeRose suggest [https://www.brookings.edu/blog/, accessed on 1 September 2022]. The majority of respondents in a study about the family formation process in Poland, where our study took place, did not find cohabitation to be an attractive option for childbearing despite its overall societal acceptance [5]. Wedlock remains the desired goal and dream living arrangement for couples. Therefore, childbearing out-of-wedlock, even when the father claims paternity, may still be considered a stressful experience for the mother, irrespective of her level of education. The results changed when the legal acknowledgment of the child by the father meant the acceptance of paternal responsibility in the unmarried status of the mother was missed. The lack of a father’s name on the birth certificate enhanced the negative effects on BWGA z-scores of children delivered by lower-educated, unmarried mothers compared to unmarried mothers with a higher education. A higher education of the mother compensated for the effect of the father’s absence. This finding supports our hypothesis that a higher educational background in single mothers creates a favorable milieu to buffer fetal growth from the negative consequences of uncertainty about the future, insecurity, and economic strain due to an enhancement in stress resilience. Therefore, we suggest treating the lack of institutional formalization of a union as an external (behavioral) factor affecting fetal growth. The educational attainment of unwed, single mothers was a significant modifier of birth outcomes when there was a lack of acknowledgment of paternity by the father. Higher-educated single mothers do not feel the same pressure associated with the economic advantages of marriage [1] because they are more likely to be economically independent, even in societies where a traditional division of labor prevails in the household. The risk of maternal stress and depression in the absence of a relationship with the baby’s father is much higher in countries where the traditional model of family still dominates [5], as compared to other Western and Northern countries [25,26].

Several studies investigated the mechanism of how education affects well-being; enhances the importance of self-confidence, self-estimation, and social interaction; and improves the chances of achieving happiness indirectly through a decent income and job prospects [27]. Better educated women usually achieve better occupational and social class positions, higher income, and other economic resources, allowing them to deal with the social stigma associated with single motherhood. A higher education is also attributed to the adoption of a healthy lifestyle, and avoidance of acknowledged risk factors such as smoking, physical stress, emotional stress, alcohol consumption, excessive weight, and low physical activity [16,22,28,29]. Education is the best indicator of socioeconomic resources that can improve economic security due to a higher chance for better employment and social status. Moreover, better-educated mothers are more likely to be employed in white-collar occupations, whereas women with only a primary education often obtain employment in lower-qualified occupations which have been associated with worse birth outcomes. Employed single mothers are happier and less stressed in parenting than single mothers who are not employed [30]. All these factors may explain the significant compensating effect of higher education on fetal growth found in unmarried mothers without father support in our study.

## 6. Conclusions

The results underline the stressful role of non-traditional relationships in affecting unfavorable neonatal outcomes. The study demonstrated that a higher level of education in mothers compensates for the negative effects of the absence of the father’s acknowledgement of the child (proxied by the missing paternal data on the birth record, which may serve as an indicator of low paternal engagement) on birth outcome, but it does not help to overcome the effect of out-of-wedlock stress exposure. Lower-educated women, whose partners offer lower or no support during pregnancy, are a vulnerable group that needs special perinatal care. We recommend addressing father support interventions in lower-educated mothers as one of the most promising areas in alleviating social risk factors and improving birth outcomes. Future research should focus on identifying psychosocial and lifestyle variables that either alone or in combination with other factors explain the differences in fetal growth.

Our conclusion would not be valid without a discussion of its limitations. Our study, similar to many studies based on birth records, is cross-sectional, and one must exercise caution when inferring causality. This is especially important when assessing birth outcomes in relation to father acknowledgement, measured by proxy indicators such as lack of paternal data on birth records, because the questionnaire may be filled in by the hospitalized mother after the child is born. Thus, the study measures associations, not consequences. Furthermore, the dataset of birth records does not allow us to assess the extent and quality of a father’s involvement during pregnancy and the type of emotional relationship between the couple, either in married or unmarried couples. We also did not know whether the current partner was the father of the baby. The dataset also lacks information on the cohabitation status among unmarried women, which is an important omission, as cohabitation is likely to be associated with a man’s willingness to invest in the child. Following a birth to an unmarried couple, cohabiting couples are much more likely to marry or remain together than non-cohabiting couples (Fragile Families Research Brief 2007). Among non-residential fathers, cohabiting men pay more child support than men that have never cohabited with the mother [8]. Due to the fact that this study was based on birth registry data, it was also impossible to include other factors that may affect our relationships, such as maternal health condition (diabetes, hypertension) or the mother’s dietary habits, or alcohol and tobacco intake before and during pregnancy.

We decided to restrict our analysis to mothers who had passed a school age (in Poland: 25 years) in order to exclude mothers whose graduate education might have been prevented or postponed by the birth. It is likely that their educational aspiration may be sometimes higher than their actual education degree. There is an obvious correlation between maternal age and education originating from the fact that some women give birth (planned or not) while studying, and because of this, some of them decide not to continue their studying. The correlation between age and education might confound the investigated relationships between father acknowledgement, education, and birth weight. After the study group restriction, we expect to see the unbiased, ”true” effect of education on birth outcome. The procedure of restricting the study group according to passing the school age while studying the effect of education on birth outcomes was also adopted by others [31].

We also decided to exclude LGA and SGA infants, because we wanted to conduct the study on homogenous neonates, without problems with fetal growth. LGA may be the result of gestational diabetes, whilst SGA indicates growth restriction. We expected that analyzing relationships between marital status, father acknowledgement, and birth outcomes on such a consistent group will strengthen our results. Finally, we noted that even in the appropriate birth-weight-for-gestational age range, statistically significant relationships were present.

Our study group may represent a very specific group of privileged women as regards marital status (91% were married), employment status (82% were classified as employed, that is having earned or unearned income), and education attainment (85% were classified as having higher education, that is passed at least final high school exams, which corresponds to British A-levels, such as upper general or specialized secondary and academic education). Due to this fact, the results are very specific for this population and the role of father recognition and mother’s education might change in other settings.

It is important to emphasize that the aim of our study was not to speak for or against a certain family structure. Instead, primary health care providers need to acknowledge the diversity and address the different needs of different families, and provide support for paternity regardless of family structure.

## Figures and Tables

**Figure 1 ijerph-20-04868-f001:**
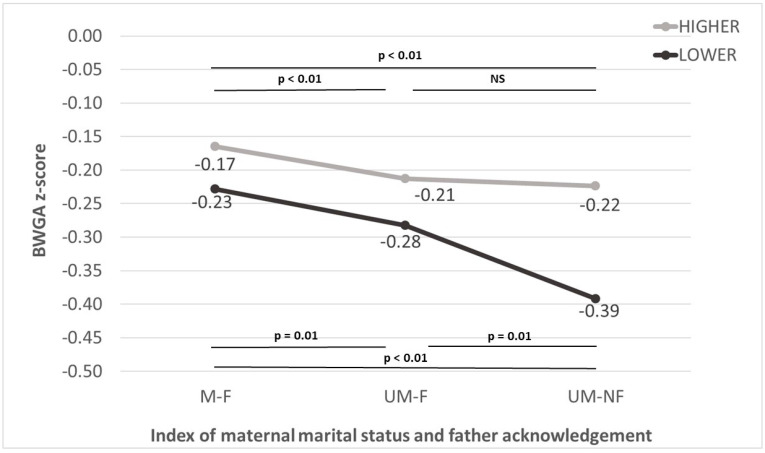
Birth-weight-for-gestational age (BWGA) z-scores in three levels of the Index of maternal marital status and father acknowledgement: married women (M-F), unmarried with father acknowledgement (UM-F), and unmarried with no father acknowledgement (UM-NF), according to lower and higher maternal education.

**Table 1 ijerph-20-04868-t001:** Characteristics of the study sample according to the Index of maternal marital status and father acknowledgement.

Characteristics	Level	Index of Maternal Marital Status and Father Acknowledgement												
		M-F				UM-F				UM-NF				
		n	%			n	%			n	%			Test, *p*-Value
Infant’s sex	girl	24,137	49.7%			1989	49.3%			503	52.9%			Pearson Chi-square: 4.1; df = 2, *p* = 0.13
	boy	24,408	50.3%			2043	50.7%			448	47.1%			
Maternal education	higher	41,950	86.5%			3087	76.9%			555	61.5%			Pearson Chi-square: 693.5; df = 2, *p* < 0.01
	lower	6521	13.5%			925	23.1%			347	38.5%			
Parity	primiparous	21,888	45.1%			2164	53.7%			513	54.0%			Pearson Chi-square: 136.0; df = 2, *p* < 0.01
	multiparous	26,655	54.9%			1868	46.3%			437	46.0%			
Maternal employment	unemployed	8073	16.7%			879	21.9%			334	37.0%			Pearson Chi-square: 316.8; df = 2, *p* < 0.01
	employed	40,390	83.3%			3130	78.1%			569	63.0%			
		**n**	**median**	**Q1**	**Q3**	**n**	**median**	**Q1**	**Q3**	**n**	**median**	**Q1**	**Q3**	
Maternal age	years	48,545	30.0	27.0	32.0	4032	30.0	27.0	34.0	951	30.0	27.0	34.0	Kruskal–Wallis test: H (2, N = 53,528) = 37.9, *p* < 0.01

M-F—married women with father acknowledgement of the child; UM-F—unmarried women with father acknowledgement of the child; UM-NF—unmarried women with no father acknowledgement of the child.

**Table 2 ijerph-20-04868-t002:** Characteristics of the study sample according to maternal education.

Characteristics	Level	Maternal Education (EDU)				Test, *p*-Value
		Higher		Lower		
		n	%	n	%	
Infant’s sex	girl	22,643	49.7%	3920	50.3%	Pearson Chi-square: 1.1; df = 1, *p* = 0.299
	boy	22,949	50.3%	3873	49.7%	
Index levels	M-F	41,950	92.0%	6521	83.7%	Pearson Chi-square: 693.5; df = 2, *p* < 0.01
	UM-F	3087	6.8%	925	11.9%	
	UM-NF	555	1.2%	347	4.5%	
Parity	primiparous	22,764	49.9%	1761	22.6%	Pearson Chi-square: 2002.3; df = 1; *p* < 0.01
	multiparous	22,826	50.1%	6032	77.4%	
Maternal employment	unemployed	6437	14.1%	2846	36.6%	Pearson Chi-square: 2329.1; df = 1, *p* < 0.01
	employed	39,130	85.9%	4937	63.4%	

EDU—maternal education; M-F—married women with father acknowledgement of the child; UM-F—unmarried women with father acknowledgement of the child; UM-NF—unmarried women with no father acknowledgement of the child.

**Table 3 ijerph-20-04868-t003:** The results of multiple linear regression with interactions, with the assumption that all predictors are unchanged, that is, set at the reference level, when being categorical (infant’s sex = girls, parity = primiparous, maternal employment status = unemployed), and for continuous—held fixed at minimum (maternal age = 25 years, year of the study = 1995). The outcome is the BWGA z-scores.

Effects		Coeff.	SE	t	*p*
**Conditional (Simple) Effects of Index Levels (UM-F vs. M-F)**					
	among higher EDU	−0.05	0.01	−4.08	<0.01
	among lower EDU	−0.05	0.02	−2.46	0.02
**Conditional (simple) effects of Index levels (UM-NF vs. M-F)**					
	among higher EDU	−0.06	0.03	−2.17	0.03
	among lower EDU	−0.16	0.03	−4.71	<0.01
**Conditional (simple) effects of Index levels (UM-NF vs. UM-F)**					
	among higher EDU	−0.01	0.03	−0.36	0.72
	among lower EDU	−0.11	0.04	−2.75	0.01
Interactions	EDU × Index (UM-F vs. M-F)	0.01	0.03	0.26	0.79
	EDU × Index (UM-NF vs. M-F)	0.10	0.04	2.39	0.02
	EDU × Index (UM-NF vs. UM-F)	−0.10	0.05	−2.00	0.05

BWGA z-scores—birth-weight-for-gestational age z-scores; EDU—maternal education; M-F—married women with father acknowledgement of the child; UM-F—unmarried women with father acknowledgement of the child; UM-NF—unmarried women with no father acknowledgement of the child. SE—standard error.

## Data Availability

Restrictions apply to the availability of these data. Data were obtained from the Central Statistical Office in Poland and are available through the data request form (https://stat.gov.pl/en/questions-and-orders/data-request-form/, accessed on 9 March 2023) with the permission of the Central Statistical Office in Poland.

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
