# Peer review of "Marital Status, Father Acknowledgement, and Birth Outcomes: Does the Maternal Education Matter?"

_ijerph, 2023, doi:10.3390/ijerph20064868_

Round 1
Reviewer 1 Report
Dear Authors,
My comments:
1. Data, which you presented are too old for me.
2. Exclusion criteria are subjective and not well-described- over 25 years? in Poland people finish 5-year studies in age 22-24, so they are high educated; Why did you exclude LGAs and SGAs and not exclude patients with chronic disease e.g. diabetes or hypertension, what will disturb you results?
3. In conclusion you write about adverse pregnancy outcomes and I did not see them in text.
4. References are too old.
Author Response
Dear Reviewer,
Thank you for your valuable suggestion and comments for our manuscript. We appreciate for your warm work earnestly and hope that the correction will meet with approval. Please find our response below.
Reviewer’s comment: „Data, which you presented are too old for me”
Our response:
The study presented in the manuscript is the first part of a broader long-term project which investigates how the changes in family formation processes affect the birth outcome in the last three decades. The analysis in this part covers the period from 1995 to 2009, which covers the beginning of the transformation of the political, economic, and social systems in Poland. The analysis of the second part, which is under study now, refers to the years 2010 - 2022.
The transformation has brought a lot of changes not only in the Polish economy but also in the openness to family diversity and the meaning of the concept of a successful life.
The manuscript evaluates the birth outcome of babies delivered by the first generation of women born and brought up during the late sixties and seventies during the communist time. Their attitude toward the traditional family as the most desirable life model has been affected by the Roman Catholic church. Therefore, only a small proportion of mothers decided to have a child out of wedlock and deliberately chose to be single mothers. However, the influence of Western culture and secularization during this 14-year period resulted in constant growth of the proportion of birth outside of marriage and reached 20% in 2009 indicating that besides the traditional family, based on marriage, more and more women were taking a decision to have and then to bring up the child on their own. This percentage has not changed much after 2009. Thus, the second part of our study will begin in 2010 and will cover the following 12 years. We expect that the results of this intended study will reflect the stronger changes in the Poles’ mentality and in particular attitudes toward family decreasing the adverse effect on birth weight newborns delivered by this new generation of mothers observed before.
Reviewer’s comment: „Exclusion criteria are subjective and not well-described- over 25 years? in Poland people finish 5-year studies in age 22-24, so they are high educated; Why did you exclude LGAs and SGAs and not exclude patients with chronic disease e.g. diabetes or hypertension, what will disturb you results?”
Our response: Thank you for this comment. We are aware that this is a true limitation of our data. But due to the fact that this study was based on birth registry data, there was impossible to include these factors, which may affect our relationships, that is maternal health condition (diabetes, hypertension) or her dietary habits, or alcohol and tobacco intake before and during pregnancy.
Therefore we included the above statement into limitations [lines 320-323].
As regards exclusion criteria, there is an obvious correlation between maternal age and education originating from the fact that some women gave birth (planned or not) while studying, and because of this, some of them decide not to continue their studying. The correlation between age and education might confound the investigated relationships between father acknowledgement, education and birth weight. We decided to restrict our analysis to mothers who has passed a school age (in Poland 25 years) in order to exclude mothers whose graduate education might have been prevented or postponed by the birth. It is likely, that their educational aspiration may be sometimes higher than their actual education degree. After the study group restriction, we expect to see the unbiased, „true” effect of education on birth outcome. The procedure of restricting the study group according to passing the school age while studying the effect of education on birth outcomes was also adopted by others, for example:
Shmueli, A., & Cullen, M. R. (1999). Birth weight, maternal age, and education: new observations from Connecticut and Virginia. The Yale journal of biology and medicine, 72(4), 245.
Moreover, while assessing the relationship between psychosocial wellbeing of mothers and birth outcomes, the common practice is to restrict the study group only to women of childbearing age (usually 25-45 years), as this is the most fertile age of woman, what was applied for example in this paper:
Wonch Hill, P., Cacciatore, J., Shreffler, K. M., & Pritchard, K. M. (2017). The loss of self: The effect of miscarriage, stillbirth, and child death on maternal self-esteem. Death Studies, 41(4), 226-235.
In addition, it seems rationale to assume that educational attainment emerges as significant predictor of birth outcome at older maternal age, primarly because being a crucial SES contribuitor.
We also decided to exclude LGA and SGA infants, because we wanted to conduct our study on homogenous neonates, without problems with fetal growth. LGA may be the result of gestational diabetes, whilst SGA indicates growth restriction. We expected, that analyzing relationships between marital status, father acknowledgement and birth outcomes on such a consisted group, will strengthen our results. Finally, we noted that even in the appropriate-birth-weight-for-gestational-age range, statistically significant relationships were present.
We described our explanation in the limitations section [lines 324-342].
Reviewer’s comment: „In conclusion you write about adverse pregnancy outcomes and I did not see them in text.”
Our response: The first statement of Conclusions states as follows: „The results underline the stressful role of non-traditional relationships in evoking adverse pregnancy outcomes”. We meant that this type of relationships may affect unfavorable neonatal outcomes, such as diminished birth weight. Therefore we re-write the sentence, and now it states as follows: „The results underline the stressful role of non-traditional relationships in affecting unfavorable neonatal outcomes.”
Reviewer’s comment: References are too old
Our response: References were up-dated.
Reviewer 2 Report
Thank you for the opportunity to review this interesting manuscript. In this study the authors used data from birth registry data to assess if birth weight for gestational age (BWGA) is associated with mother civil status and father recognition, and if so the role of mother´s education level among more than 50,000 Polish mothers resident in Krakow between 1995-2009. Overall, this study contributes to the literature by deepening our understanding of how family and couples’ structure and education may affect child´s birth outcomes.
Findings are generally consistent with the current knowledge of effect of mother´s level of education on birth outcomes.
Overall, the manuscript is well written and informative. The methods and statistical analysis were conducted properly. There are some comments below aimed to improve its quality for publication in the journal.
-The authors restricted the sample to singleton born at term (37-42 weeks) from mothers above 25 years old with exclusion of large and small babies for gestational age (LGA, SGA). Excluding SGA and LGA might include a selection bias and therefore I suggest including a sensitivity analysis analyzing all live birth at term.
- The results showed that most mothers were married (91%), employed (82%) and with higher education attainment (85%) which represent a very specific group of privileged women when compared to characteristics or education and employment for women in other countries. I recommend the authors to comment on this topic in the discussion session as the results are very specific for this population and the role of father recognition and mother´s education might change in other settings.
Author Response
Dear Reviewer,
Thank you for your valuable suggestion and comments for our manuscript. We appreciate for your warm work earnestly and hope that the correction will meet with approval. Please find our response below.
Reviewer’s comment: The authors restricted the sample to singleton born at term (37-42 weeks) from mothers above 25 years old with exclusion of large and small babies for gestational age (LGA, SGA). Excluding SGA and LGA might include a selection bias and therefore I suggest including a sensitivity analysis analyzing all live birth at term.
Our response:
Thank you for this comment. We included all births born at term, that is between 37-42 weeks of gestational age, but excluded LGA and SGA infants, because we wanted to conduct our study on homogenous neonates, without problems with fetal growth. LGA may be the result of gestational diabetes, whilst SGA indicates growth restriction. We expected, that analyzing relationships between marital status, father acknowledgement and birth outcomes on such a consisted group, will strengthen our results. Finally, we noted that even in the appropriate-birth-weight-for-gestational-age range, statistically significant relationships were present.
We described our explanation in the limitations section [lines 336-342].
Reviewer’s comment: "The results showed that most mothers were married (91%), employed (82%) and with higher education attainment (85%) which represent a very specific group of privileged women when compared to characteristics or education and employment for women in other countries. I recommend the authors to comment on this topic in the discussion session as the results are very specific for this population and the role of father recognition and mother´s education might change in other settings."
Our response:
Thank you for this remark. It is true, that our study group may represent a very specific group of privileged women as regards marital status (91% were married), employment status (82% were classified as employed that is having earned or unearned income) and education attainment (85% were classified as having higher education, that is passed at least final high school exams, which corresponds to British A-levels, such as: upper general or specialized secondary and academic education). Due to this fact the results are very specific for this population and the role of father recognition and mother´s education might change in other settings. It would be interesting to investigate similar patterns in other countries.
We added this explanation into limitations [lines 341-360].
Round 2
Reviewer 1 Report
Dear Authors,
I accept you reply.